# Nomograms for Predicting the Risk and Prognosis of Liver Metastases in Pancreatic Cancer: A Population-Based Analysis

**DOI:** 10.3390/jpm13030409

**Published:** 2023-02-24

**Authors:** Huaqing Shi, Xin Li, Zhou Chen, Wenkai Jiang, Shi Dong, Ru He, Wence Zhou

**Affiliations:** 1Second College of Clinical Medicine, Lanzhou University, Lanzhou 730000, China; 2The First Clinical Medical College, Lanzhou University, Lanzhou 730030, China; 3Department of General Surgery, Lanzhou University Second Hospital, Lanzhou 730030, China

**Keywords:** nomogram, pancreatic cancer, liver metastases, predictive models, overall survival, SEER database

## Abstract

The liver is the most prevalent location of distant metastasis for pancreatic cancer (PC), which is highly aggressive. Pancreatic cancer with liver metastases (PCLM) patients have a poor prognosis. Furthermore, there is a lack of effective predictive tools for anticipating the diagnostic and prognostic techniques that are needed for the PCLM patients in current clinical work. Therefore, we aimed to construct two nomogram predictive models incorporating common clinical indicators to anticipate the risk factors and prognosis for PCLM patients. Clinicopathological information on pancreatic cancer that referred to patients who had been diagnosed between the years of 2004 and 2015 was extracted from the Surveillance, Epidemiology, and End Results (SEER) database. Univariate and multivariate logistic regression analyses and a Cox regression analysis were utilized to recognize the independent risk variables and independent predictive factors for the PCLM patients, respectively. Using the independent risk as well as prognostic factors derived from the multivariate regression analysis, we constructed two novel nomogram models for predicting the risk and prognosis of PCLM patients. The area under the curve (AUC) of the receiver operating characteristic (ROC) curve, the consistency index (C-index), and the calibration curve were then utilized to establish the accuracy of the nomograms’ predictions and their discriminability between groups. Using a decision curve analysis (DCA), the clinical values of the two predictors were examined. Finally, we utilized Kaplan–Meier curves to examine the effects of different factors on the prognostic overall survival (OS). As many as 1898 PCLM patients were screened. The patient’s sex, primary site, histopathological type, grade, T stage, N stage, bone metastases, lung metastases, tumor size, surgical resection, radiotherapy, and chemotherapy were all found to be independent risks variables for PCLM in a multivariate logistic regression analysis. Using a multivariate Cox regression analysis, we discovered that age, histopathological type, grade, bone metastasis, lung metastasis, tumor size, and surgery were all independent prognostic variables for PCLM. According to these factors, two nomogram models were developed to anticipate the prognostic OS as well as the risk variables for the progression of PCLM in PCLM patients, and a web-based version of the prediction model was constructed. The diagnostic nomogram model had a C-index of 0.884 (95% CI: 0.876–0.892); the prognostic model had a C-index of 0.686 (95% CI: 0.648–0.722) in the training cohort and a C-index of 0.705 (95% CI: 0.647–0.758) in the validation cohort. Subsequent AUC, calibration curve, and DCA analyses revealed that the risk and predictive model of PCLM had high accuracy as well as efficacy for clinical application. The nomograms constructed can effectively predict risk and prognosis factors in PCLM patients, which facilitates personalized clinical decision-making for patients.

## 1. Introduction

Pancreatic cancer is among the most fatal malignancies, and patients have a poor prognosis [1,2]. According to Cancer Statistics 2022, PC is the fourth biggest cause of cancer-related fatalities, following lung, prostate, breast, and colorectal cancers [3]. It is predicted that PC will become the third biggest cause of cancer mortalities in European countries by 2025 [4]. Because the pancreas is located in the retroperitoneum, its location is deep, the early symptoms of PC are not obvious, the opportunity for early diagnosis and treatment is often missed, and the five-year survival rate is below 10% [5,6,7]. Surgery is currently the only curative method available, but only about 20% of patients are suitable for radical surgery, with a five-year survival rate of about 21% following surgery; in the two years following surgery, over 80% of patients will have a recurrence or metastasis [8,9,10]. Distant metastases are the main cause of poor prognoses in PC patients, with a systemic metastasis rate of more than 50% [11,12]. The median survival of patients with distant metastases from untreated PC is less than 6 months, and the combination of adjuvant chemotherapy with gemcitabine and albumin paclitaxel remains as the first-line treatment option in clinical practice [13,14].

The liver is the most frequent site of distant metastasis in PC, followed by the peritoneum and lung [15,16]. In a study involving 13,233 patients with distant metastases from pancreatic cancer, Oweira et al. found that liver metastases accounted for 76% of PC patients [17]. It has been found that surgical resection, radiotherapy, and radiofrequency ablation of the liver are feasible for improving the prognosis of PCLM patients according to their liver metastases [18,19,20,21]. The TNM staging method of the American Joint Committee on Cancer (AJCC) has been utilized to assess the prognosis of pancreatic cancer patients; there is a lack of comprehensive and effective models for anticipating and assessing the risk factors and prognosis of PCLM patients [22]. Although some artificial intelligence (AI) models have been applied in the medical field, there are limitations in terms of hardware, computing power, and data [23,24]. We used the relatively convenient R software to build nomogram models that were specifically designed for the study of PCLM patients. Therefore, clinically relevant information needs to be integrated to determine the risk factors and prognosis of liver metastasis in PC patients. Previous studies have constructed nomogram models of distant metastases in pancreatic cancer patients, including those for bone metastases and lung metastases, to predict risk factors and prognosis, respectively [11,25,26]. However, predictive models for liver metastasis in PC patients have not been adequately studied.

The nomogram prediction model outperforms the traditional TNM staging system [27]. A nomogram can integrate multiple clinical factors and personalize the patient’s situation to accurately and effectively assess the causes and factors affecting the prognosis. [28,29]. Nomogram prediction models have been widely used in the prognosis and risk assessment of various tumors such as gastric, colorectal, prostate, breast, and liver cancers [30,31,32,33,34]; its visual interface and web-based model facilitate practical use by clinicians and patients, facilitating individualized patient treatment [35]. Currently, no nomogram model exists to anticipate the risk and prognosis in PCLM patients. Therefore, we used a large sample of the SEER database’s clinical information, combined with commonly used clinical indicators, to construct two nomograms that anticipate the risk and prognosis of patients suffering from PCLM; in addition, we also web-based the models for clinical use.

## 2. Methods

### 2.1. Data Sources and Extraction

PC patient data from 2004 to 2015 were extracted from the SEER database for this investigation. About 47.9% of U.S. residents is covered by the SEER database, which is a widely-used and dependable public cancer database [36]. The clinicopathological data of PC patients were extracted from the SEER database (Database name = Incidence − SEER Research Plus Data, 18 Registries, Nov 2020 Sub (2000–2018)—Linked To County Attributes—Total U.S., 1969–2019 Counties) utilizing SEER*Stat 8.4.0 (http://seer.cancer.gov (accessed on 17 August 2022)). The patient information used from the SEER database is public, anonymous, and does not require signed informed permission or approval from an ethics review board. The account number for this authorization is 15071-Nov2021. The inclusion criteria for the PC patients in this study were as follows: (1) histopathologically confirmed diagnosis of pancreatic cancer; (2) tumor primary site was the pancreas; (3) complete clinicopathological information was available; and (4) follow-up information was complete and valid. Exclusion criteria were: (1) patients with a survival time of 0; (2) unknown surgical status; (3) patients with a pathological type other than pancreatic cancer; and (4) case data from cases obtained by autopsy and retention of death reports only. Finally, a total of 12,327 pancreatic cancer patients were enrolled, of which 1898 had liver metastases.

### 2.2. Nomogram Construction and Validation

In our study, variables such as: age; sex; race; main site; histological type grade; T stage; N stage; tumor size; metastases to the brain, bone, and lungs; and treatment with surgery, radiation, and chemotherapy were utilized to recognize risk and predictive variables for liver metastases pancreatic cancer patients. All patients formed a diagnostic cohort for the risk factor analysis and for the construction of diagnostic models. In a 7:3 ratio, PCLM patients were randomized into two groups, a “training cohort” and a “validation cohort”. In the training cohort, independent predictive variables for PCLM patients were evaluated utilizing univariate and multivariate Cox regression analyses, and nomograms were created to anticipate the PCLM patients’ OS at 1, 3, and 5 years. We then used the C-index, AUC, and calibration curves to analyze the discriminatory ability and accuracy of the models, and we used the DCA to examine the clinical efficacy of the two models. OS was the major result for predicting survival in this study, which was identified as the period from the date of diagnosis until the date of death or latest follow-up.

### 2.3. Statistical Analysis

In this study, categorical factors were reported as integers and percentages, and factors were compared utilizing chi-square tests or Fisher exact tests. In the first diagnostic cohort, univariate and multivariate logistic regression analyses were conducted and multivariate logistic analyses were carried out to calculate the odds ratio (OR) and the matching 95% confidence interval (CI) for every distinct risk variable. The training cohort was then subjected to univariate and multivariate Cox regression analyses, and multivariate Cox regression analysis was used to obtain independent predictive variables.

On the basis of these independent risk and predictive variables, two nomograms were subsequently developed to anticipate risk and prognosis in PCLM patients. In the training and validation cohorts, PCLM patients were split into high- and low-risk groups due to median risk scores, and the potential differences in OS between these predictive variables were compared using Kaplan–Meier curves. Statistical analyses and model construction were conducted utilizing the SPSS 22.0 tool (IBM, Chicago, IL, USA) and R tool (version 4.0.3) (https://www.r-project.org/ (accessed on 17 August 2022)); the results were judged as statistically significant if *p* < 0.05.

## 3. Results

### 3.1. Baseline Characteristics of Patients

Based on the screening criteria, we screened a total of 12,327 PC patients from the SEER database, of which 1898 patients had liver metastases. According to Table 1, we found that age was not statistically significant in patients with and without LM (*p* > 0.05), and the number of patients between the ages of 50 and 64 years as well as the number between 65 and 74 years was similar (34.8% vs. 32.7%). There were more male patients (51.1%) among all patients, and they were predominantly white (78.5%). The main location of the tumor was at the pancreatic head in 57.5% of patients, followed by the tail of the body of the pancreas in 28.8% of patients. The histopathological type was predominantly adenocarcinoma (53.2%), of which 42.5% were moderately differentiated. Patients with T3 staging accounted for 59.5% of the sample, followed by T2 accounting for 18%. N stage typing accounted for 50% of each total case, but N0 accounted for 55.9% of LM patients. Regarding the patients, 98.9% were without bone metastases, 99.9% were without brain metastases, and 96.5% were without lung metastases. The tumor size was predominantly 2–4 cm (48.1%), followed by a size of >4 cm accounting for 31.2% of patients. In terms of treatment, surgical patients accounted for 69.9% of the sample and chemotherapy patients accounted for 59.7%; however, radiotherapy patients were less common, accounting for 22.9% of the sample.

### 3.2. Independent Risk Factors of PCLM

The univariate logistic analysis of 15 common clinical indicators (age, sex, race, primary site, histological subtype, grading, T stage, N stage, bone metastasis, brain metastasis, lung metastasis, tumor size, surgery, radiotherapy, and chemotherapy) revealed that age was not statistically significant (*p* > 0.05). A subsequent multivariate logistic regression analysis revealed that sex, primary site, histological subtype, grade, T stage, N stage, bone metastasis, lung metastasis, tumor size, surgery, radiotherapy, and chemotherapy were independent risk variables for PCLM (Table 2).

### 3.3. Diagnostic Nomogram Construction and Validation

On the basis of the independent risk variables obtained from a multivariate logistic regression analysis of the diagnostic cohort, we constructed a nomogram for predicting PCLM risk factors (Figure 1) and web-based the model at the following link: https://shqycmx.shinyapps.io/Diagnostic_Model/ (accessed on 17 August 2022). The C-index of the model was 0.884 (95% CI: 0.876–0.892), and the AUC value was 0.872. By observing the calibration curve, it was found that the reported findings closely matched the anticipated findings, and these results indicated that the model possessed a greater level of prediction accuracy and discrimination capability. The DCA analysis revealed that the model was more selective than the old TNM staging approach used in clinical practice (Figure 2).

### 3.4. Independent Prognostic Factors of PCLM

At a 7:3 ratio, 1898 PCLM patients were randomized into a training cohort and validation cohort, and chi-square and Fisher exact tests established no significant variations between all variables in the two cohorts (Table 3, *p* > 0.05). A univariate Cox regression analysis of the training cohort demonstrated that age, histopathological type, tumor differentiation degree, T stage, bone metastasis, lung metastasis, tumor size, and surgery were statistically significant variables (*p* < 0.05). Age, histological type, degree of tumor differentiation, bone metastases, lung metastases, tumor size, and surgical treatment were independent predictive variables for the predictive OS in PCLM patients, as determined by a multivariate Cox regression analysis (Table 4).

### 3.5. Predictive Nomogram Construction and Validation

On the basis of the independent predictive variables obtained by the multivariate Cox regression analysis in the training cohort, we created a predictive model for PCLM patients (Figure 3), which was also web-based at the following link: https://shqycmx.shinyapps.io/Prognostic_model/ (accessed on 17 August 2022). In the training cohort, the AUC values for the prediction of the 1-year, 3-year, and 5-year OS were 0.764, 0.903, and 0.937, respectively. In the validation cohort, the AUC values were 0.783, 0.909, and 0.937, respectively (Figure 4). It was found that as the prediction time became longer, the prediction capability of the model became stronger. The C-index and calibration curves were utilized to validate the prediction power of the predictive model. The nomogram’s C-index was 0.686 (95% CI: 0.648–0.722) and 0.705 (95% CI: 0.647–0.758) in the training cohort and validation cohort, respectively. Observing the calibration curves at 1, 3, and 5 years for both cohorts found a substantial correlation between the actual data and the model’s predictions (Figure 5). The DCA results showed that the model exhibited a significantly good net gain in terms of mortality risk and outperformed the TNM staging method, showing high clinical utility for anticipating OS in PCLM patients (Figure 6). The Kaplan–Meier survival analysis by independent prognostic factors and high- and low-risk cohorts revealed significant differences in the survival between subgroups (Figure 7, *p* < 0.05).

## 4. Discussion

Pancreatic cancer, especially pancreatic ductal adenocarcinoma (which accounts for more than 90% of pancreatic cancer), is a severe aggressive tumor of the digestive system. About 80% of pancreatic cancer is already locally advanced or has distant metastases when detected [37,38]. The blood supply of the pancreas also determines that the liver is the most prevalent location of distant pancreatic metastasis. The pancreatic head mostly obtains its blood supply from branches of the gastroduodenal and superior mesenteric arteries, whereas the tail of the pancreas’ body obtains its blood supply primarily from splenic artery branches. The pancreatic venous blood flow eventually enters the liver through the portal vein, and when pancreatic cancer invades the blood vessels, the shed cells tend to colonize the liver with the blood flow [39,40]. In the last decade, the prognosis of PC has not improved as much as other tumors with the advancement of medical technology [41]. The prevalence of PC remains at 3%, but the mortality rate has increased from 6% in 2012 to 8% in 2022 [3,42]. There is evidence that for patients with PCLM who respond well to chemotherapy and are physically able to tolerate surgery, surgery after chemotherapy may be beneficial [43,44]. However, in PC patients with multiple liver metastases, the opportunity for surgery is often lost, and this is an important reason for the poor prognosis of PCLM patients [45]. Therefore, we need to identify risk and predictive variables for liver metastasis in patients with PC, facilitate early detection and prevention, and assess the prognosis of PCLM patients. In this paper, by constructing the diagnostic and prognostic nomograms of PCLM, we are able to calculate the diagnostic- and prognostic-related scores by combining the actual conditions of the patients and by providing guidance for individualized patient treatment.

We know that artificial intelligence is currently being used in many areas, including in healthcare. Deep learning is an important branch of artificial intelligence [46]. Deep learning evolved from the study of artificial neural networks; however, it is not identical to conventional neural networks. Nevertheless, in terms of vocabulary, the many deep learning algorithms, including deep reinforcement learning, generative adversarial networks, recurrent neural networks, and convolutional neural networks, use the phrase “neural network” [47,48,49,50]. Deep learning can be thought of as a semi-theoretical, semi-empirical modelling approach that employs human understanding of mathematics and computer algorithms, along with as much training information as is possible, to construct an architectural framework, utilizing the massive computing power of computers to tune the internal criteria to approximate the issue’s objectives as closely as possible [51]. There are many advantages to predictive models built using deep learning over our nomogram predictive models: deep learning technology performs very well and is very capable of learning; the neural networks of deep learning have many layers and are so wide that they can theoretically map to arbitrary functions so that they can overcome very difficult issues [52]; deep learning is extremely reliant on data, and more data are better for its performance; platform compatibility is high, etc. [53]. Similarly, the disadvantages of deep learning when compared with nomogram prediction models are prominent. Deep learning demands large amounts of data and computer power, making it expensive. In addition, numerous programs are not yet compatible with mobile devices. Deep learning demands a great level of computational power, and standard CPUs can no longer match its requirements, such that the hardware requirements are high and so are the costs. The model design for deep learning demands a substantial commitment of time and resources to build new algorithms and models, so it is highly complex [47]. Most people can only use off-the-shelf models. As deep learning is data-dependent and is not very interpretable, it is easy to have bias, etc. [54]. In summary, we chose to use R software to process the SEER data and construct the nomograms prediction model.

Few articles have researched PCLM risk factors alone. In gastrointestinal tumors such as liver, gastric, and colorectal cancers, the risk of concomitant liver metastases was significantly associated with advanced age, pathological type, poor tumor differentiation, bone metastases, lung metastases, high TNM stage, and unsystematic treatment [55,56,57]. In this investigation, we found that the risk variables for PC patients with concomitant liver metastasis were: male, main tumor site, pathological type, grade, T2 stage, N1 stage, metastases in the lungs and bones, larger tumor, and chemotherapy, which was a similar outcome to the results of previous studies [25]. Previous research has found that radiotherapy is a risk variable for bone metastases from pancreatic cancer; however, in our study, we found that radiotherapy was actually beneficial for patients with liver metastases [26]. This is probably because, on the one hand, the liver is relatively located on a superficial location, which facilitates stereotactic radiotherapy including 3D conformal radiotherapy techniques; on the other hand, pancreatic cancer has dense fibrous connective tissue, and this characteristic limits drug penetration to produce drug resistance while relatively limiting the tumor spread. Radiotherapy with radioactive particle implantation may be beneficial for patients with PCLM [58,59,60,61,62,63,64,65]. Based on these risk factors for liver metastasis, prompt CT or MRI imaging is clinically indicated for patients with suspected PCLM. We found by the multivariate Cox analysis that the age, type of pathology, degree of tumor differentiation, surgery, tumor size, bone metastasis, and lung metastasis were independent predictive variables of LM in PC patients. Combined with the risk and predictive variables that were acquired herein, the prognostic OS of PCLM patients can be more effectively assessed clinically to further guide the clinical decision-making and evaluation of PCBM patients.

In our study, we found that the main age group of PCLM patients was between 50 and 74 years. According to the Kaplan–Meier survival analysis, the prognosis of older patients had worse OS compared with younger patients [66,67]. This confirms earlier findings that older patients could be more susceptible to tumor immune escape as their autoimmune capacity decreases [68]. Epithelial–mesenchymal transition is closely related to tumor immune escape and can cause immune escape and distant metastasis by inducing the suppression of the CD8+ T cell role and by creating a local tumor suppressive microenvironment [69,70]. In the U.S., PC occurrence is slightly higher in men than in women, and in our study, we found more male patients with PCLM, accounting for 55.6% of the sample, which could be linked to men’s bad diet and lifestyle habits; however, the specific mechanism needs further study [71].

It has been reported that 60–70% of PC occurs in the head of the pancreas and about 15% each in the tail or the body of the pancreas [72]. In this study, the primary tumor site being in the tail of the pancreatic body was a risk factor for liver metastasis. This is related to the anatomical site of the pancreatic body. Besides invading the portal vein, pancreatic body tumors also tend to locally invade the liver and superior mesenteric vessels; meanwhile, besides invading the splenic vein, the growth of pancreatic tail tumors is usually unimpeded due to fewer adjacent structures [73]. Adenocarcinoma is the most common type of pathology in PC and is the one with the worst prognosis [74]. This is consistent with our findings, and the best relative prognosis for neuroendocrine cancer was found by survival analysis. In our study, the degree of tumor differentiation, bone metastases, brain metastases, and tumor size were risk and predictive variables for PCLM. Moreover, surgery was determined to be beneficial for patients with PCLM, whereas chemotherapy and radiotherapy had few results in terms of improving the prognosis of patients with advanced disease [75,76].

We used the clinical information of a large sample from the SEER database to construct a relatively complete and valid comprehensive evaluation model that could effectively assess the risk and outcome of PCLM patients. This study elucidated the function of multiple risk variables in estimating the outcome of PCLM patients. Constructing a web-based nomogram facilitates the actual use of the model, which is easy for clinicians to evaluate and make decisions based on the actual condition of patients; additionally, it also facilitates patients to actively cooperate with the treatment according to their prognosis. However, this research has certain drawbacks: first, it has some selection bias due to its retrospective nature, and subsequent prospective studies are needed; second, some clinical treatment information, such as tumor markers, is currently missing from the SEER database, and subsequent external validation is needed; finally, the clinical data are mainly from the U.S. population, and global multicenter investigations are needed for validating and improving the model applicability.

## 5. Conclusions

In this study, we identified risk variables for PCLM using univariate and multivariate logistic regression analyses and predictive factors for PCLM utilizing univariate and multivariate Cox regression analyses to establish two nomograms to anticipate the risk and outcome of PCLM. We then built a web-based nomogram that allows clinicians to facilitate early diagnosis, select personalized therapy strategies for PCLM patients, and improve patient prognosis.

## Figures and Tables

**Figure 1 jpm-13-00409-f001:**
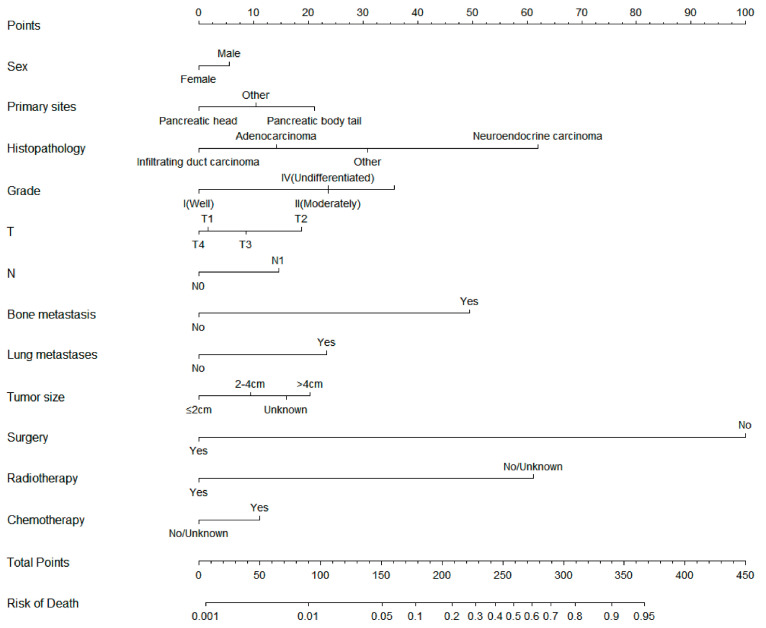
Diagnostic nomogram for predicting the risk of liver metastasis in patients with pancreatic cancer.

**Figure 2 jpm-13-00409-f002:**
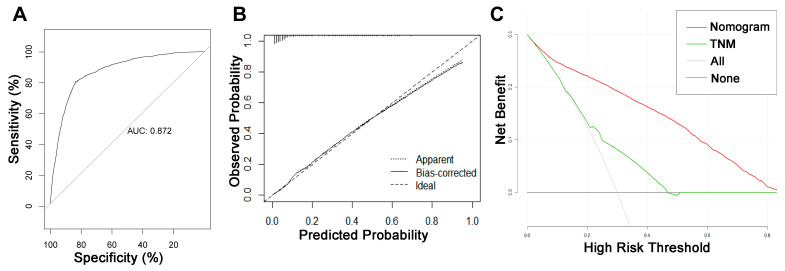
ROC curve, calibration curve, and decision curve analysis of the nomogram model for the risk of liver metastasis in pancreatic cancer. (**A**) AUC values were used to evaluate the predictive performance of the nomogram. (**B**) Calibration curves of the nomogram model. The diagonal 45-degree line indicates a perfect prediction. (**C**) Comparison of the diagnostic nomograms and TNM staging DCA curves. The net benefit, calculated by adding the true positives minus the false positives, corresponds to the measurement on the Y-axis; the X-axis represents the threshold probability.

**Figure 3 jpm-13-00409-f003:**
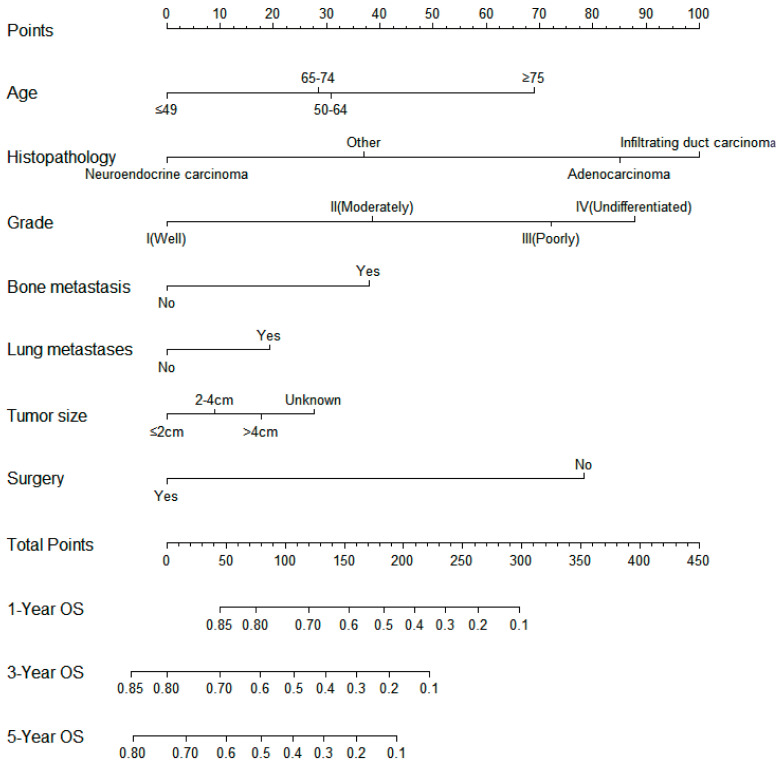
Nomogram model for predicting the overall survival in patients with liver metastases from pancreatic cancer. The scale corresponds to the variable axis according to the patient variables. The overall probability of patient survival at 1, 3, and 5 years can be assessed by drawing a vertical line upward to determine the score for each variable corresponding to the point where the sum of these point scores lies on the total point axis and by drawing a vertical line downward on the survival axis.

**Figure 4 jpm-13-00409-f004:**
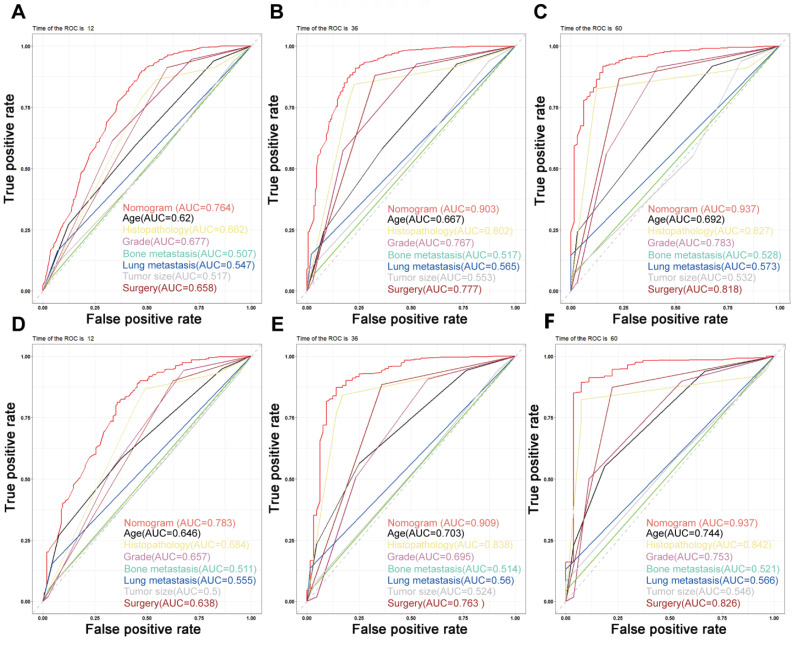
Time-dependent ROC curves. (**A**–**C**) Time-dependent ROC curves for the prognostic nomogram show AUC values of 0.764, 0.903, and 0.937 for predicting the 1-year, 3-year, and 5-year overall survival in the training cohort, respectively. (**D**–**F**) The AUC values were 0.783, 0.909, and 0.937 for predicting the 1-, 3-, and 5-year overall survival in the validation cohort, respectively.

**Figure 5 jpm-13-00409-f005:**
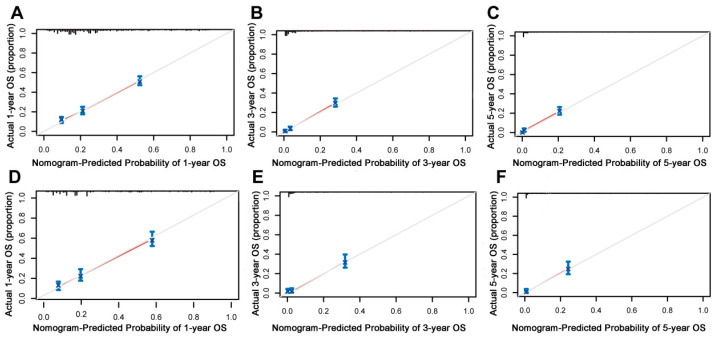
Calibration curves of the prognostic nomogram model. (**A**–**C**) Calibration curves for the 1-year, 3-year, and 5-year overall survival of patients with liver metastases from pancreatic cancer in the training cohort and (**D**–**F**) validation cohort. The black line represents the ideal reference line, and the blue line is calculated by bootstrapping (resampling: 1000) and represents the nomogram performance. The closer the solid red line is to the black line, the more accurate the model is at predicting overall survival.

**Figure 6 jpm-13-00409-f006:**
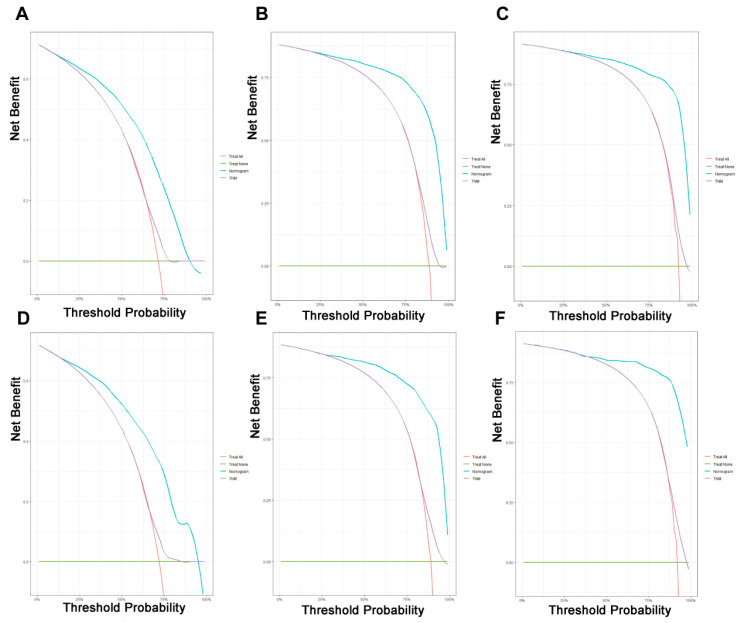
Decision curve analysis of the prognostic nomogram and TNM staging system for predicting the overall survival in patients with liver metastases from pancreatic cancer. (**A**–**C**) Survival benefits at 1, 3, and 5 years in the training cohort. (**D**–**F**) Survival benefits at 1, 3, and 5 years in the validation cohort.

**Figure 7 jpm-13-00409-f007:**
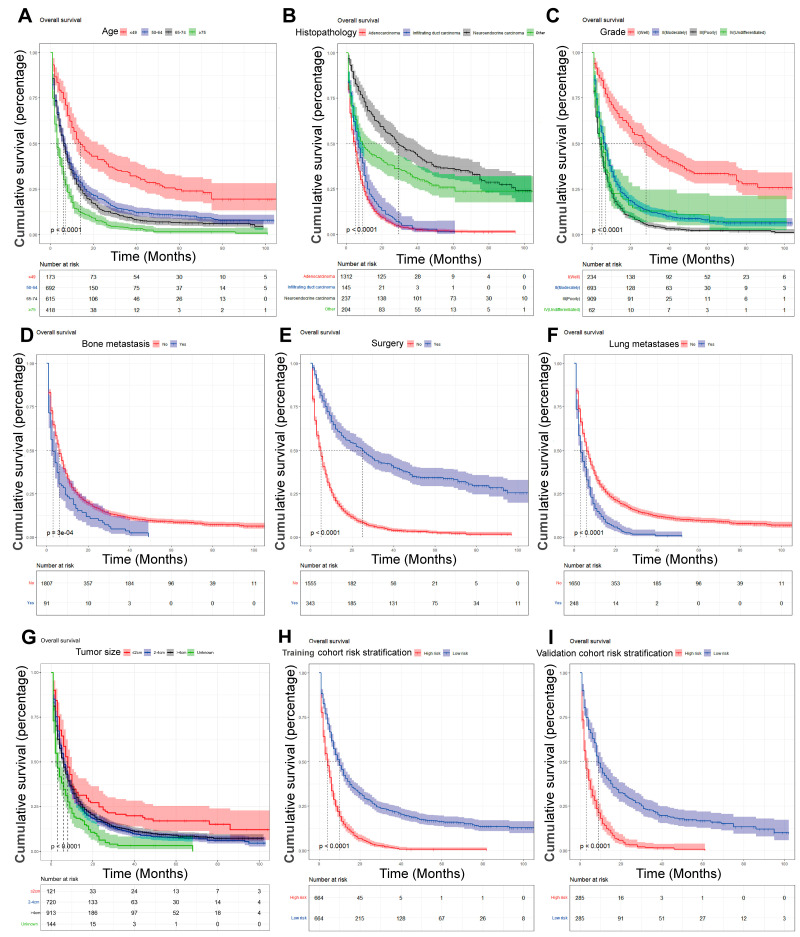
Kaplan–Meier analysis of the prognostic overall survival in patients with liver metastases from pancreatic cancer. (**A**) Age; (**B**) histopathology; (**C**) grade; (**D**) bone metastasis; (**E**) surgery; (**F**) lung metastasis; (**G**) tumor size; (**H**) train cohort risk stratification; and (**I**) validation cohort risk stratification.

**Table 1 jpm-13-00409-t001:** Baseline clinical features and treatment regimens in 12,327 patients with pancreatic cancer.

Variables	*n* (%)	Without LM Cohort *n* (%)	With LM Cohort *n* (%)	*p*
Age	12,327	10,429	1898	
≤49	1043 (8.5)	870 (8.3)	173 (9.1)	0.088
50–64	4294 (34.8)	3602 (34.5)	692 (36.5)	
65–74	4032 (32.7)	3417 (32.8)	615 (32.4)	
≥75	2958 (24.0)	2540 (24.4)	418 (22.0)	
Sex				
Female	6029 (48.9)	5187 (49.7)	842 (44.4)	<0.001
Male	6298 (51.1)	5242 (50.3)	1056 (55.6)	
Race				
Black	1586 (12.9)	1281 (12.3)	305 (16.1)	<0.001
Other	1066 (8.6)	921 (8.8)	145 (7.6)	
White	9675 (78.5)	8227 (78.9)	1448 (76.3)	
Primary sites				
Other	1686 (13.7)	1320 (12.7)	366 (19.3)	<0.001
Pancreatic body tail	3552 (28.8)	2820 (27.0)	732 (38.6)	
Pancreatic head	7089 (57.5)	6289 (60.3)	800 (42.1)	
Histopathology				
Adenocarcinoma	6555 (53.2)	5243 (50.3)	1312 (69.1)	<0.001
Infiltrating duct carcinoma	3173 (25.7)	3028 (29.0)	145 (7.6)	
Neuroendocrine carcinoma	1160 (9.4)	923 (8.9)	237 (12.5)	
Other	1439 (11.7)	1235 (11.8)	204 (10.7)	
Grade				
I (Well)	2673 (21.7)	2439 (23.4)	234 (12.3)	<0.001
II (Moderately)	5244 (42.5)	4551 (43.6)	693 (36.5)	
III (Poorly)	4172 (33.8)	3263 (31.3)	909 (47.9)	
IV (Undifferentiated)	238 (1.9)	176 (1.7)	62 (3.3)	
T stage				
T1	1246 (10.1)	1172 (11.2)	74 (3.9)	<0.001
T2	2223 (18.0)	1648 (15.8)	575 (30.3)	
T3	7331 (59.5)	6521 (62.5)	810 (42.7)	
T4	1527 (12.4)	1088 (10.4)	439 (23.1)	
N stage				
N0	6158 (50.0)	5097 (48.9)	1061 (55.9)	<0.001
N1	6169 (50.0)	5332 (51.1)	837 (44.1)	
Bone metastasis				
No	12,194 (98.9)	10,387 (99.6)	1807 (95.2)	<0.001
Yes	133 (1.1)	42 (0.4)	91 (4.8)	
Brain metastasis				
No	12,316 (99.9)	10,423 (99.9)	1893 (99.7)	0.019
Yes	11 (0.1)	6 (0.1)	5 (0.3)	
Lung metastasis				
No	11,895 (96.5)	10,245 (98.2)	1650 (86.9)	<0.001
Yes	432 (3.5)	184 (1.8)	248 (13.1)	
Tumor size				
≤2 cm	2099 (17.0)	1978 (19.0)	121 (6.4)	<0.001
2–4 cm	5933 (48.1)	5213 (50.0)	720 (37.9)	
>4 cm	3848 (31.2)	2935 (28.1)	913 (48.1)	
Unknown	447 (3.6)	303 (2.9)	144 (7.6)	
Surgery				
No	3708 (30.1)	2153 (20.6)	1555 (81.9)	<0.001
Yes	8619 (69.9)	8276 (79.4)	343 (18.1)	
Radiotherapy				
None/Unknown	9502 (77.1)	7725 (74.1)	1777 (93.6)	<0.001
Yes	2825 (22.9)	2704 (25.9)	121 (6.4)	
Chemotherapy				
None/Unknown	4972 (40.3)	4270 (40.9)	702 (37.0)	0.001
Yes	7355 (59.7)	6159 (59.1)	1196 (63.0)	

**Table 2 jpm-13-00409-t002:** Univariate and multivariate logistic regression analysis of the risk factors in patients of pancreatic cancer with liver metastases.

	Univariate Analysis			Multivariate Analysis		
Variables	OR	95% CI	*p*	OR	95% CI	*p*
Age						
≤49	Ref					
50–64	0.966	0.805–1.159	0.711			
65–74	0.905	0.753–1.088	0.289			
≥75	0.828	0.682–1.004	0.055			
Sex						
Female	Ref					
Male	1.241	1.125–1.369	0	1.171	1.037–1.322	0.011
Race						
Black	Ref					
Other	0.661	0.532–0.818	0	0.804	0.618–1.045	0.103
White	0.739	0.646–0.849	0	0.875	0.739–1.036	0.121
Primary sites						
Other	Ref					
Pancreatic body tail	0.936	0.813–1.079	0.361	1.338	1.121–1.597	0.001
Pancreatic head	0.459	0.400–0.527	0	0.751	0.633–0.890	0.001
Histopathology						
Adenocarcinoma	Ref					
Infiltrating duct carcinoma	0.191	0.160–0.228	0	0.676	0.550–0.832	0
Neuroendocrine carcinoma	1.026	0.877–1.196	0.745	3.69	2.931–4.645	0
Other	0.660	0.561–0.773	0	1.576	1.261–1.970	0
Grade						
I (Well)	Ref					
II (Moderately)	1.587	1.360–1.859	0	1.903	1.557–2.325	0
III (Poorly)	2.904	2.496–3.390	0	2.652	2.170–3.240	0
IV (Undifferentiated)	3.672	2.654–5.028	0	1.918	1.278–2.879	0.002
T stage						
T1	Ref					
T2	5.526	4.317–7.171	0	1.591	1.019–2.484	0.041
T3	1.967	1.549–2.535	0	1.207	0.785–1.858	0.391
T4	6.390	4.959–8.343	0	0.955	0.610–1.495	0.841
N stage						
N0	Ref					
N1	0.754	0.683–0.832	0	1.49	1.307–1.698	0
Bone metastasis						
No	Ref					
Yes	12.454	8.672–18.179	0	3.982	2.552–6.212	0
Brain metastasis						
No	Ref					
Yes	4.588	1.321–15.250	0.012	0.569	0.146–2.214	0.416
Lung metastasis						
No	Ref					
Yes	8.369	6.873–10.208	0	1.907	1.515–2.400	0
Tumor size						
≤2 cm	Ref					
2–4 cm	2.258	1.857–2.768	0	1.292	0.914–1.827	0.147
>4 cm	5.085	4.189–6.226	0	1.734	1.226–2.452	0.002
Unknown	7.769	5.934–10.192	0	1.537	1.028–2.298	0.036
Surgery						
No	Ref					
Yes	0.057	0.051–0.065	0	0.066	0.056–0.078	0
Radiotherapy						
None/Unknown	Ref					
Yes	0.195	0.160–0.234	0	0.190	0.153–0.236	0
Chemotherapy						
None/Unknown	Ref					
Yes	1.181	1.068–1.307	0.001	1.359	1.190–1.551	0

OR, Odds ratio; CI, confidence intervals.

**Table 3 jpm-13-00409-t003:** Demographic and clinicopathologic characteristics of 1898 patients of pancreatic cancer with liver metastases.

Variables	*n* (%)	Training Cohort *n* (%)	Validation Cohort *n* (%)	*p*
Age	1898	1328	570	
≤49	173 (9.1)	128 (9.6)	45 (7.9)	0.176
50–64	692 (36.5)	464 (34.9)	228 (40.0)	
65–74	615 (32.4)	438 (33.0)	177 (31.1)	
≥75	418 (22.0)	298 (22.4)	120 (21.1)	
Sex				
Female	842 (44.4)	572 (43.1)	270 (47.4)	0.094
Male	1056 (55.6)	756 (56.9)	300 (52.6)	
Race				
Black	305 (16.1)	204 (15.4)	101 (17.7)	0.426
Other	145 (7.6)	101 (7.6)	44 (7.7)	
White	1448 (76.3)	1023 (77.0)	425 (74.6)	
Primary sites				
Other	366 (19.3)	264 (19.9)	102 (17.9)	0.224
Pancreatic body tail	732 (38.6)	521 (39.2)	211 (37.0)	
Pancreatic head	800 (42.1)	543 (40.9)	257 (45.1)	
Histopathology				
Adenocarcinoma	1312 (69.1)	919 (69.2)	393 (68.9)	0.493
Infiltrating duct carcinoma	145 (7.6)	105 (7.9)	40 (7.0)	
Neuroendocrine carcinoma	237 (12.5)	157 (11.8)	80 (14.0)	
Other	204 (10.7)	147 (11.1)	57 (10.0)	
Grade				
I (Well)	234 (12.3)	157 (11.8)	77 (13.5)	0.061
II (Moderately)	693 (36.5)	469 (35.3)	224 (39.3)	
III (Poorly)	909 (47.9)	652 (49.1)	257 (45.1)	
IV (Undifferentiated)	62 (3.3)	50 (3.8)	12 (2.1)	
T stage				
T1	74 (3.9)	52 (3.9)	22 (3.9)	0.897
T2	575 (30.3)	397 (29.9)	178 (31.2)	
T3	810 (42.7)	574 (43.2)	236 (41.4)	
T4	439 (23.1)	305 (23.0)	134 (23.5)	
N stage				
N0	1061 (55.9)	745 (56.1)	316 (55.4)	0.829
N1	837 (44.1)	583 (43.9)	254 (44.6)	
Bone metastasis				
No	1807 (95.2)	1260 (94.9)	547 (96.0)	0.37
Yes	91 (4.8)	68 (5.1)	23 (4.0)	
Brain metastasis				
No	1893 (99.7)	1324 (99.7)	569 (99.8)	0.999
Yes	5 (0.3)	4 (0.3)	1 (0.2)	
Lung metastasis				
No	1650 (86.9)	1150 (86.6)	500 (87.7)	0.554
Yes	248 (13.1)	178 (13.4)	70 (12.3)	
Tumor size				
≤2 cm	121 (6.4)	89 (6.7)	32 (5.6)	0.682
2–4 cm	720 (37.9)	510 (38.4)	210 (36.8)	
>4 cm	913 (48.1)	630 (47.4)	283 (49.6)	
Unknown	144 (7.6)	99 (7.5)	45 (7.9)	
Surgery				
No	1555 (81.9)	1087 (81.9)	468 (82.1)	0.947
Yes	343 (18.1)	241 (18.1)	102 (17.9)	
Radiotherapy				
None/Unknown	1777 (93.6)	1238 (93.2)	539 (94.6)	0.321
Yes	121 (6.4)	90 (6.8)	31 (5.4)	
Chemotherapy				
None/Unknown	702 (37.0)	472 (35.5)	230 (40.4)	0.053
Yes	1196 (63.0)	856 (64.5)	340 (59.6)	

**Table 4 jpm-13-00409-t004:** Univariate and multivariate Cox regression models for overall survival in patients of pancreatic cancer with liver metastases.

	Univariate Analysis			Multivariate Analysis		
Variables	HR	95% CI	*p*	HR	95% CI	*p*
Age						
≤49	Ref					
50–64	1.822	1.462–2.27	0	1.37	1.09–1.71	0.006
65–74	1.916	1.536–2.389	0	1.34	1.07–1.68	0.011
≥75	2.913	2.313–3.67	0	2.06	1.63–2.61	0
Sex						
Female	Ref					
Male	1.004	0.896–1.125	0.942			
Race						
Black	Ref					
Other	0.94	0.729–1.211	0.63			
White	0.978	0.836–1.146	0.786			
Primary sites						
Other	Ref					
Pancreatic body tail	0.92	0.786–1.076	0.294			
Pancreatic head	1.062	0.91–1.239	0.446			
Histopathology						
Adenocarcinoma	Ref					
Infiltrating duct carcinoma	0.791	0.644–0.97	0.025	1.2	0.97–1.5	0.092
Neuroendocrine carcinoma	0.251	0.204–0.31	0	0.4	0.32–0.51	0
Other	0.402	0.328–0.494	0	0.59	0.48–0.74	0
Grade						
I (Well)	Ref					
II (Moderately)	2.62	2.108–3.255	0	1.51	1.19–1.9	0.001
III (Poorly)	3.957	3.198–4.896	0	2.18	1.73–2.74	0
IV (Undifferentiated)	3.178	2.241–4.505	0	2.62	1.84–3.73	0
T stage						
T1	Ref					
T2	1.667	1.206–2.303	0.002	0.9	0.56–1.45	0.664
T3	1.264	0.919–1.738	0.149	0.76	0.48–1.2	0.241
T4	1.755	1.264–2.435	0.001	0.8	0.5–1.28	0.349
N stage						
N0	Ref					
N1	0.909	0.812–1.019	0.102			
Bone metastasis						
No	Ref					
Yes	1.481	1.157–1.895	0.002	1.51	1.17–1.94	0.001
Brain metastasis						
No	Ref					
Yes	1.834	0.687–4.897	0.226			
Lung metastasis						
No	Ref					
Yes	1.72	1.464–2.02	0	1.22	1.04–1.44	0.016
Tumor size						
≤2 cm	Ref					
2–4 cm	1.428	1.117–1.825	0.005	1.18	0.83–1.7	0.356
>4 cm	1.392	1.092–1.774	0.007	1.32	0.92–1.88	0.126
Unknown	2.168	1.6–2.938	0	1.53	1.03–2.29	0.036
Surgery						
No	Ref					
Yes	0.289	0.244–0.343	0	0.46	0.38–0.55	0
Radiotherapy						
None/Unknown	Ref					
Yes	0.852	0.682–1.065	0.16			
Chemotherapy						
None/Unknown	Ref					
Yes	0.989	0.875–1.117	0.856			

HR, hazard ratio. CI, confidence intervals.

## Data Availability

The raw data supporting the conclusions of this article will be made available by the authors without undue reservation.

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
