# Peer review of "Nomograms for Predicting the Risk and Prognosis of Liver Metastases in Pancreatic Cancer: A Population-Based Analysis"

_jpm, 2023, doi:10.3390/jpm13030409_

Round 1

Reviewer 1 Report

Some points need to be revised:

- Abstract is too long. Please reduce it.

- ".. to construct two nomograms predicting the risk and prognosis of PCLM patients, and web-based them for clinical use." What is the reason for making 2 monograms? Improve the outcome ? Better follow-up ?

- "In this paper, by constructing the diagnostic nomogram and prognostic nomogram of...  conditions of patients and providing guidance for individualized patient treatment." What do authors mean ?

- In discussion section: "which facilitates stereotactic radiotherapy including 3D conformal radiotherapy techniques..." Improve this part reporting the role of 3D and VR in surgery. Consider these important papers: -- PMCID: PMC9141435 -- DOI: 10.3390/ijerph19106347 -- PMCID: PMC7314924 -- DOI: 10.3389/fsurg.2020.00038 -- PMID: 24967996 -- DOI: 10.1055/s-0034-1368573

- Figure 4 should be enlarged as writing is difficult to read. Revide it.

- Conclusion contains little news. What does this paper add new compare to previous papers?

Author Response

  1. Abstract is too long. Please reduce it.

Revised, Lines 23 to 49.

We have streamlined the content of the abstract and the specific changes can be found in the manuscript.

  1. ".. to construct two nomograms predicting the risk and prognosis of PCLM patients, and web-based them for clinical use." What is the reason for making 2 monograms? Improve the outcome ? Better follow-up ?

Revised,

Thank you very much for your valuable comments. The prognosis of pancreatic cancer patients is still mainly assessed by the TNM staging system, and common prognostic indicators such as tumour site and pathological type are not included in the assessment, so it is not possible to make a comprehensive and effective assessment. Our nomogram model, which incorporates common clinical indicators, enables a more comprehensive and accurate assessment of patient prognosis and risk factors, and the application of this model enables personalised clinical decision making for patients, allowing the content and duration of follow-up to be adjusted according to the patient's specific condition, effectively improving patient prognosis.

  1. "In this paper, by constructing the diagnostic nomogram and prognostic nomogram of...  conditions of patients and providing guidance for individualized patient treatment." What do authors mean ?"

Revised,

In this study, regression analyses were performed on age, gender, race, tumour site, pathological type, degree of tumour differentiation, stage, common distant metastatic sites, tumour size and treatment, and models were constructed to predict prognosis and risk factors respectively based on the results of the multivariate regression analysis. For example, according to the diagnostic nomogram model, bone metastases are found to be a risk factor for pancreatic cancer with liver metastases, so for pancreatic cancer patients with bone metastases, we should pay attention to liver examination when we examine or follow up patients so that liver metastases can be detected and treated in time.For example, some patients and families in clinical practice refuse surgery, believing it to be dangerous. Based on the prognostic nomogram model, we found that the prognosis was better for patients who were treated surgically, so for patients with an indication for surgery, the combination of the model in the preoperative conversation would be more convincing in predicting the outcome.

  1. In discussion section: "which facilitates stereotactic radiotherapy including 3D conformal radiotherapy techniques..." Improve this part reporting the role of 3D and VR in surgery. Consider these important papers: -- PMCID: PMC9141435 -- DOI: 10.3390/ijerph19106347 -- PMCID: PMC7314924 -- DOI: 10.3389/fsurg.2020.00038 -- PMID: 24967996 -- DOI: 10.1055/s-0034-1368573.

Revised,

Thank you for taking the time to provide us with these fruitful and insightful articles, which we have referenced in the discussion section.

  1. Figure 4 should be enlarged as writing is difficult to read. Revide it.

Revised,

Figure 4 has been revised in the manuscript.

  1. Conclusion contains little news. What does this paper add new compare to previous papers?

Revised,

Compared to the conventional TNM staging system, we have incorporated more clinical predictors to make the predictions more reliable. In contrast to previous studies on pancreatic cancer, this study specifically focuses on patients with liver metastases from pancreatic cancer, while constructing models to predict prognosis and risk factors, and webpage them for practical clinical use.

Reviewer 2 Report

General comments:

This is a highly topical and scientifically thorough paper that deals with nomogram-based predictions of liver metastases risks among patients diagnosed with pancreatic cancer. The scientific approach is adequately described, and the paper is generally well written, though the writing style could improve (certain phrases are too long and difficult to follow).

The following are my comments to further improve the manuscript:

(1)   Introduction: Nomogram-based predictions are only compared to the traditional TNM staging system. However, there are several predictive models using various artificial intelligence tools, such as deep learning, that showed promising results in this direction. Please include a paragraph in the Introduction presenting this aspect.

(2)   Also, the Discussion section should include a paragraph on the advantages/ shortcomings of nomogram-based predictive models as compared to deep learning-based models.

Specific comments:

1.     Line 18 – “to predict the diagnosis and prediction...” – please reword

2.     Line 48 – “…and were web-based.’ – this section does not fit in the sentence

3.     Line 111 – ‘In our study, variables such as age,…’ (remove ‘these’)

4.     Lines 227-228 – ‘However, we found that radiotherapy is a risk factor in bone metastases from pancreatic 227 cancer, but in our study, we found that radiotherapy is practically beneficial..” please rephrase to make a clear difference between literature findings and the results of your study. Also, divide this phrase into 2 sentences as it is hard to follow in its current format.

5.     Lines 239-240 – ‘In our study, we found that the main age group of PCLM patients was between 50-74 years, which is also consistent with the age group of pancreatic cancer prevalence…’  - of course it is, as your results are based on the same statistics / database. The first part of this phrase is rather redundant, and I suggest its removal.

6.     Line 248 – ‘…and the specific mechanism needs further study’ reword to “however, the specific mechanisms need further studies.”

7.     Line 263 – ‘Also, a research gap has been filled for the piece of the comprehensive prediction model for PCLM patients.’ – please reformulate into a lighter statement; ‘This study elucidated the role of various risk factors in predicting the outcome of PCLM patients’ (or something along these lines).

Author Response

(1) Introduction: Nomogram-based predictions are only compared to the traditional TNM staging system. However, there are several predictive models using various artificial intelligence tools, such as deep learning, that showed promising results in this direction. Please include a paragraph in the Introduction presenting this aspect.

Revised, Lines 76 to 78.

We have added the relevant content to the introduction, as follows. Although some artificial intelligence (AI) models have been applied to the medical field, there are limitations in terms of hardware, computing power and data. We use the relatively convenient R software to build nomograms models specifically for the study of PCLM patients.

(2)   Also, the Discussion section should include a paragraph on the advantages/ shortcomings of nomogram-based predictive models as compared to deep learning-based models.

Revised, Lines 224 to 245.

We have added relevant deep learning content to the discussion section, as described in the discussion section of the manuscript.

Specific comments:

  1. Line 18 – “to predict the diagnosis and prediction...” – please reword

Revised,

Thank you very much for taking the time to give us detailed instructions on how to revise our manuscript. We have made changes in the manuscript.

  1. Line 48 – “…and were web-based.’ – this section does not fit in the sentence

Revised,

Revisions have been made and are detailed in the manuscript.

  1. Line 111 – ‘In our study, variables such as age,…’ (remove ‘these’)

Revised,

Already revised.

  1. Lines 227-228 – ‘However, we found that radiotherapy is a risk factor in bone metastases from pancreatic 227 cancer, but in our study, we found that radiotherapy is practically beneficial..” please rephrase to make a clear difference between literature findings and the results of your study. Also, divide this phrase into 2 sentences as it is hard to follow in its current format.

Revised,

Revised, see manuscript for details.

  1. Lines 239-240 – ‘In our study, we found that the main age group of PCLM patients was between 50-74 years, which is also consistent with the age group of pancreatic cancer prevalence…’  - of course it is, as your results are based on the same statistics / database. The first part of this phrase is rather redundant, and I suggest its removal.

Revised,

Repetitive statements have been revised, see the manuscript for details.

  1. Line 248 – ‘…and the specific mechanism needs further study’ reword to “however, the specific mechanisms need further studies.”

Revised,

Thank you for your careful guidance, which has been revised in the text.

  1. Line 263 – ‘Also, a research gap has been filled for the piece of the comprehensive prediction model for PCLM patients.’ – please reformulate into a lighter statement; ‘This study elucidated the role of various risk factors in predicting the outcome of PCLM patients’ (or something along these lines).

Revised,

Changes have been made in the text and are detailed in the manuscript.

Reviewer 3 Report

The authors submitted an interesting original article dealing with a population-based analyses and the monogram for predicting risk and prognosis of liver metastases from pancreatic cancer. Since the pancreatic cancer is a serious diagnose, which shows increasing incidence in population of European, Asian and North American countries, the topic of this manuscript is important and timely.

The manuscript is well-prepared, and the text shows a logic structure. For the investigation, the authors use adequate methods. The data support the results and conclusions. Additionally, the manuscript is accompanied with large supporting information and unpublished materials. Since I found no serious weaknesses, I recommend the manuscript for publication.

Author Response

Dear Professor,

We have seen your review comments on our articles. Thank you very much for your acknowledgement and support of our research.

Best wishes,

Huaqing Shi

Round 2

Reviewer 1 Report

Authors solved all my criticisms.